# Diversity and Spread of Acetolactate Synthase Allelic Variants at Position 574 Endowing Resistance in *Amaranthus hybridus* in Italy

**DOI:** 10.3390/plants12020332

**Published:** 2023-01-10

**Authors:** Andrea Milani, Silvia Panozzo, Samuele Pinton, Renato Antonio Danielis, Maurizio Sattin, Laura Scarabel

**Affiliations:** 1Institute for Sustainable Plant Protection (IPSP-CNR), 35020 Legnaro, Italy; 2Regional Agency for Rural Development of Friuli Venezia Giulia (ERSA), 33050 Pozzuolo del Friuli, Italy

**Keywords:** acetolactate synthase resistance, smooth pigweed, point mutations, alleles, single-seed DNA extraction, resistance management

## Abstract

Poor control of *Amaranthus* spp. with herbicides inhibiting acetolactate synthase (ALS) has been observed for several years in soybean fields in north-eastern Italy, but to date only a few ALS-resistant populations have been confirmed. An extensive sampling of putatively resistant *Amaranthus* accessions was completed in the Friuli Venezia Giulia region, across an arable land area of about 3000 km^2^. In total, 58 accessions were tested to confirm their resistance status, recognize the *Amaranthus* species, identify the mutant *ALS* alleles endowing the resistance and determine the efficacy of 3 pre-emergence herbicides. Most accessions resulted in cross-resistance to thifensulfuron-methyl and imazamox. Genomic DNA were extracted from single seeds with a newly developed protocol; an allele-specific PCR assay revealed the presence of the 574-leucine in 20 accessions, of the 574-methionine in 22, and of both alleles in 9 accessions. The two variants showed a different spatial distribution. All resistant populations were ascribed to *A. hybridus*. *A. hybridus* resistant to ALS herbicides is well-established in this Italian region and its resistance is due to two ALS mutant alleles. Metribuzin, clomazone and metobromuron can be used as alternative herbicides to be applied in pre-emergence and they should be integrated into the management strategies to limit the spread of resistance.

## 1. Introduction

Weeds are among the pests that can determine crop losses if not adequately controlled [1]. Herbicides remain, by far, the main strategy adopted to manage them, even if it is strongly recommended to integrate them with other management tactics [2]. The repeated use of herbicides can have negative consequences and lead to the evolution of resistant weed populations, threatening crop sustainability.

The *Amaranthus* genus is frequently involved in herbicide resistance: until now, 9 species (out of the roughly 70 commonly recognized) have evolved resistance to at least one herbicide site of action (SoA) [3]. Dioecious species appear to be more prone to evolve resistance, as their outcrossing nature favors the recombination of resistant alleles among accessions. To date, the dioecious species *Amaranthus palmeri* S. Watson and *Amaranthus tuberculatus* (Moq.) J.D. Sauer have evolved resistance up to 7 [3] and 6 [4] herbicide sites of action, respectively. Among the monoecious, *Amaranthus hybridus* L. has evolved resistance up to 5 and *Amaranthus retroflexus* L. up to 3 SoA [3].

*Amaranthus* species are all troublesome weeds in summer crops such as soybean, cotton, maize, carrots, tomatoes and potatoes [5] and acetolactate synthase (ALS) inhibiting herbicides are commonly used for their chemical control properties. Worldwide, this herbicide group (HRAC group 2) including six chemical families (sulfonylureas, imidazolinones, triazolinones, triazolopyrimidines, pyrimidinyl benzoates and sulfonanilides) is frequently used, as they have high efficacy at low doses against many broadleaf weeds and are selective for major crops [6].

Resistance to ALS inhibitors in dicot weeds is mainly due to point mutations at *ALS*, the gene encoding the target protein of the herbicide. To date, in *Amaranthus* species, amino acid substitutions endowing ALS resistance have been identified in seven different positions of the *ALS* gene: Ala-122, Pro-197, Ala-205, Asp-376, Trp-574, Ser-653 and Gly-654 [3]. The tryptophan-574-leucine substitution is very well-described and has often been reported to confer cross-resistance to different chemical families within the ALS inhibitors. A second possible substitution at this site is the 574-methionine, which has been observed only once in *Apera spica-venti* in 2014 [7] and once in *A. hybridus* in 2017 [8], appearing as isolated cases.

Reports of ALS-resistant *Amaranthus* biotypes are increasing in Europe [8,9,10,11]. In Italy, they have been recorded for all four above-mentioned *Amaranthus* species, and the resistance was due to point mutations at the *ALS* gene. Soybean is the most affected crop and field populations sometimes appear to be a mix of two or more species. *A. retroflexus*, *A. hybridus* and *A. tuberculatus* were found to infest a field at the same time in 2017 [8], while ALS-resistant *A. palmeri* was found in 2018 [12] living sympatrically with *A. hybridus* and *A. retroflexus* (Milani, unpublished data). Since different species appear to have a different risk of evolving complex resistance traits, correct identification is crucial for the integrated management of this pest. Soybean is cultivated extensively in north-eastern Italy and the high reliance on ALS inhibitors to control broadleaved weeds led to the appearance of the first ALS-resistant *A. hybridus* biotype in 2003 [13]. Despite several farmers informally reporting increasing control failures in the last 20 years in this area, only one more resistant case was confirmed in 2012 (but the resistance mechanism was not investigated) [14]. Therefore, the real diffusion of ALS resistance was believed to be largely underestimated. To overcome this paucity of information, in 2018, the phytosanitary service of the Regional Agency for Rural Development of Friuli Venezia Giulia (ERSA) organized a detailed monitoring of most areas where soybean is cultivated in the region. Fifty-seven herbicide-treated soybean fields mainly infested by *Amaranthus* spp. were found, and resistance to ALS inhibitors was suspected.

The aims of this study were (a) to determine the occurrence of *Amaranthus* resistant to ALS inhibitors across the Italian region of Friuli Venezia Giulia, (b) to determine the *Amaranthus* species involved, (c) to determine the main resistance mechanism and (d) to evaluate the susceptibility to non-ALS herbicides.

## 2. Results

### 2.1. Pattern of Resistance to ALS Inhibitors

No statistical differences were recorded in the survival rate and VEB values between the two experiments, therefore the data were pooled and analyzed. The susceptible check (accession 17–53) was completely controlled by both thifensulfuron-methyl and imazamox. Since the survival rates against thifensulfuron-methyl and imazamox were very similar, only the results referring to thifensulfuron-methyl are reported. Most of the 58 tested accessions were not adequately controlled: 48 had survival rates higher than 90% and 3 had rates between 25% and 90% (Appendix A). Also, the VEB values were very similar between the two herbicide treatments and generally near those of the untreated control, indicating that the resistance mechanism was very likely to have been target-site mediated. Since the first recorded ALS-resistant biotype in this region had a mutation at codon 574 of *ALS* [13] and was later ascribed to *A. hybridus* [8], the presence of resistance-endowing alleles at this locus was investigated through PCR and Sanger sequencing after species identification. Subsequent analyses were focused on the 51 populations having survival rates against thifensulfuron-methyl higher than 25% (Appendix A).

### 2.2. Amaranthus Species Identification

Analysis of the images (51 of the whole inflorescences, e.g., Figure 1A, 510 of the single inflorescences, e.g., Figure 1B and 204 of the flowers and seeds, e.g., Figure 1C) suggested that all the accessions belong to the same species, *A. hybridus*. All the samples were clearly monoecious. Looking at the female flowers, the membranous border of the bracts was interrupted at the mid-flower bract (Figure 1B): this characteristic excluded most of the species of the *A. hybridus* aggregate, limiting the possible species to *A. hybridus* or *A. cruentus*. Since the bracts were generally 1.5–2 times longer than the perianth and the tepals were very similar to each other (actinomorphic flowers, Figure 1C), all the accessions were ascribed to *A. hybridus*.

### 2.3. Resistance Mechanism

#### 2.3.1. DNA Extraction from Single Seeds

The DNA concentration ranged from 7.30 to 21.40 ng µL^−1^, while the A260/280 absorbance ratio ranged from 1.60 to 2.42 and the A260/230 absorbance ratio ranged from 0.74 to 1.56. The DNA extracted from single seeds were successfully used for *ALS* partial amplification and the allele-specific PCR assay.

#### 2.3.2. ALS Partial Amplification and Sequencing

To look for the point mutations responsible for ALS resistance, gDNA was extracted from the seeds of accessions 20, 34, 42, 45, 49 and 56 (3 seeds each). The partial sequencing of *ALS* revealed that three allelic variants of codon 574 of the *ALS* gene were present among the six tested accessions, distributed across the sampling area: the wild type, coding for tryptophan (Trp), and two resistance-endowing variants, one encoding for leucine (Leu) and one for methionine (Met). An extract of the chromatograms is reported as a Appendix A. In accessions 42, 49 and 56, only the Leu allele was found, while in accession 34, only Met was found. Instead, in accession 20, Met was found in 2 seeds and Leu in 1 seed. In accession 45, Leu was found in 2 seeds and 1 seed had both Leu and Trp.

#### 2.3.3. Allele-Specific PCR Assay (PASA) Development and Application

The allele-specific PCR assay resulted in being highly specific in identifying samples with the mutant (leucine and methionine) or the wild type (tryptophan) alleles (Figure 2). Even if a band is visible in one seed that resulted in being positive to tryptophan when only methionine was expected (sample n 4, Figure 2), it should be considered as an aspecific amplification and ignored because it is very weak compared to the specific bands.

After the set-up, the PASA was used to detect leucine and methionine alleles among the 204 seed samples (4 seeds for each of the 51 accessions; see Appendix A). A single amplification band was detected in 94 and 93 seed samples out of 204, for leucine and methionine, respectively. Nine seed samples gave no amplification and were assumed to be wild type. Eight seed samples resulted in being positive to both leucine and methionine. Among the 51 accessions, at least 2 out of the 4 tested seeds for each accession had a mutant allele: 20 accessions had the leucine allele, while 22 had the methionine and 9 had both (meaning they had seeds that were positive for both alleles or a mix of seeds with only one allele) (Figure 3). An example of the PASA gel electrophoresis and a summary of the results are reported in Appendix A, respectively.

### 2.4. Susceptibility to Pre-Emergence Herbicides

The same six accessions described in Section 2.3.2 (namely, 20, 34, 42, 45, 49 and 56, plus the susceptible check 17–53) were tested with non-ALS herbicides applied in pre-emergence. The results of the two experiments were pooled because the t-test detected no differences between repetitions (*p* < 0.05). The efficacy of metribuzin was 100% for all accessions, whereas that of metobromuron ranged from 98% to 100% (Figure 4). The fresh weight of the very few plants surviving the metobromuron was 0.2% with respect to the untreated control for accession 20, and less than 0.1% for accessions 45 and 56. The efficacy of clomazone ranged from 95% to 100% for most accessions, except for no. 42 (70%) (Figure 4). Despite the efficacy being lower than that observed for the other accessions, the fresh weight of the survivors was 5.6% compared to the untreated controls.

## 3. Discussion

This is the first regional-scale study reporting the occurrence of *Amaranthus* spp. ALS resistance in soybean fields across north-eastern Italy, and one of the few in Europe. Previous regional-scale studies in Europe focused on ALS-resistant barnyard grass (*Echinochloa* spp.) [17,18], ALS-resistant common ragweed (*Ambrosia artemisiifolia* L.) [19] and ALS or ACCase-resistant blackgrass (*Alopecurus myosuroides* Huds.) [20,21].

Most of the tested accessions proved to be ALS-resistant, showing that a huge area of Friuli Venezia Giulia, where soybean is a very important crop, is affected by ALS-resistance. Through the analysis of flower characteristics, all resistant plants were ascribed to *A. hybridus*. While, in the nearby Veneto region, several *Amaranthus* species were found in the same field [8], in the Friuli Venezia Giulia region, infestations of dioecious *Amaranthus* species (i.e., *A. tuberculatus* and *A. palmeri*) have not yet been reported. This is a favorable situation for two reasons: the monoecious species have a lower risk of evolving multiple resistance compared to the dioecious and uniform populations may have a more similar germination time.

A very fast, room temperature and liquid nitrogen-free, non-toxic protocol of DNA extraction from *Amaranthus* seeds was set up, and it can hasten the detection of resistance-endowing alleles in routine screenings. The method is suitable for PCR amplification as well as for Sanger sequencing. A limit of the method is that two different genomes are extracted at the same time because, in angiosperms, the embryo (and the endosperm) develop within maternal tissues (typically teguments and/or the pericarp). For this reason, for example, a seed sample that is positive for two alleles cannot be called heterozygous, i.e., the genotype cannot be determined, because the maternal and the embryo tissues are extracted together. Furthermore, nucleotide polymorphisms in the proximity of the 3′ end of a primer could drastically affect the amplification efficiency and thus, for example, detecting a single mutant allele does not mean that all the extracted tissues are homozygous for that mutation. Despite this limit, detecting a resistance allele in a seed sample is already proof that the population is evolving resistance, even if the allele was in the maternal tissue, because *ALS* mutations are fully dominant and always inherited and expressed by the progeny [22]. *A. hybridus* is a diploid (2n = 32) [23], mostly self-pollinated species (a 10% outcrossing rate is estimated) [24], thus, if a mother plant is heterozygous for a resistance allele, the 75% of its progeny will be resistant to ALS inhibitors.

The ALS-resistance mechanism in all the investigated accessions is target-site mediated, confirming what has been previously observed for this genus [25,26]. Up to now, all records of ALS-resistant amaranths were due to point mutations at *ALS*, except for a population of *A. tuberculatus* that evolved nontarget-site resistance [27]. Furthermore, the observed cross-resistance pattern clearly indicated that the resistance was due to a point mutation at position 574 of *ALS*. For these reasons, *ALS* was only partially sequenced to highlight the allelic variants at this locus and no other resistance mechanisms were investigated.

Two allelic variants endowing resistance were found at position 574 of *ALS* and they appeared to not be uniformly distributed: the substitution 574-methionine is concentrated in the center and south-east of the region, while leucine is concentrated in the north and west (Figure 3), and the reason is not known. The 574-leucine substitution has often been reported to confer broad cross-resistance to ALS inhibitors; instead, the 574-methionine substitution is not common, and has only been reported in *Apera spica-venti* [7] and in the previously mentioned Italian *A. hybridus* biotype found in the nearby Veneto region [8], collected about 30 km away from the area investigated here. Notably, accession 58, sampled in 2012, had the methionine allele, meaning that this allele was already present in this area for quite a long time.

The methionine allele has rarely been observed in other weed species, likely because the tryptophan-574-methionine substitution requires a double nucleotide change (TGG → ATG), which is a very rare event with respect to the single nucleotide change necessary for the tryptophan-574-leucine substitution (TGG → TTG). Alternatively, the methionine allele might have evolved from the leucine after a single nucleotide change (TTG → ATG). In this latter case, to be fixed in the population and replace the leucine allele, the methionine allele should have an evolutionary advantage (i.e., lower fitness cost or a higher inhibition constant in the presence of ALS inhibitors). The presence of the methionine allele in southern populations might be associated with the different weed management strategy of the double-cropping practice (soybean after wheat) that is very common across the whole region. Because of the pedo-climatic conditions, pre-emergence herbicides are widely used in the northern area, while the lower summer rainfall in the southern area discourages the use of those herbicides. Therefore, weed control in the southern area solely relies on ALS-inhibiting herbicides that exert very high selection pressure.

The efficacy of the metribuzin and metobromuron applied in pre-emergence to control these ALS-resistant *A. hybridus* populations was nearly 100%, therefore these herbicides can be used alone to control these accessions, as previously observed with other *Amaranthus* species [8,12,28]. The efficacy of clomazone was slightly lower than the other two herbicides. In keeping with our result, other authors reported that clomazone efficacy on *A. hybridus* decreases 12 days after application [29]. Since *Amaranthus* spp. are indicated as “moderately susceptible” in most clomazone-based products, they should not be used alone, and only when their activity against specific weeds is required (e.g., versus *Abutilon theophrasti* Medik.). The residual activity of pre-emergence herbicides is a relevant characteristic as the emergence of *Amaranthus* seedlings may be scattered over several weeks. The integration of pre-emergence herbicides in a weed management program is recommended to allow for a proper rotation of the herbicide SoA and therefore a reduction of the selective pressure on the weeds.

Within the European Union, Italy is the first soybean producer: in 2020, 260 thousand hectares were harvested [30], yielding 3.9 t ha^−1^, while in 2022, the harvesting area increased to 360 thousand hectares [31]. In Europe, the adoption of restrictive legislations (e.g., Directive 2009/128/EC on the sustainable use of pesticides, Regulation (EC) No. 1107/2009 and the European Water Framework Directive 2000/60/EC) led to a remarkable reduction in the number of herbicides available on the market. If not properly managed, the remaining molecules may exert an increasing selective pressure on the weeds. The discovery and fast confirmation of new resistance cases is the key to limit their expansion, aiming to maintain the efficacy of herbicides and the sustainability of soybean production.

## 4. Materials and Methods

### 4.1. Sampling and Plant Material

Since the area to be monitored was very large, the sampling was conducted along a pre-planned route (route-based approach), by exploring localities where soybean was known to be cultivated. Virtually, all soybean fields in the region (north-eastern Italy) are treated with ALS inhibitors (imazamox + thifensulfuron-methyl), since this is the most common weed control practice in that area. In total, 57 putatively ALS-resistant *Amaranthus* spp. accessions were collected in 2018 (named 1 to 57) from fields with very dense infestations of *Amaranthus* spp., and at least 20 plants per field were sampled. The ALS-resistant population collected in 2012 [14], with unknown resistance mechanism and species (accession 58), plus a herbicide-susceptible reference accession (17–53, *A. hybridus*) were also included. The geographical distribution of the sampled accessions is reported in Figure 5.

### 4.2. Pattern of Resistance to ALS Inhibitors

An herbicide treatment with thifensulfuron-methyl and imazamox was carried out to assess the resistance status of the accessions and select those with high survival rate to study the main resistance mechanism involved. The bioassay was performed twice in a greenhouse located in north-eastern Italy (45°21′ N, 11°58′ E) where temperatures varied between 15 and 20 °C and from 25 to 34 °C, during night and day, respectively. Seeds were sown in agarose medium, vernalized and then transplanted in standard potting mix (20 seedlings per pot/replicate, with two replicates per each accession). At three- to four-leaf stage (13–14 of the BBCH scale [32]), plants were treated with thifensulfuron-methyl at the recommended field rate of 6 g a.i. ha^−1^ (Harmony 50 SX, Corteva™, 500 g a.i. 1 kg^−1^) or imazamox at 40 g a.i. ha^−1^ (Tuareg^®^, Corteva 40 g a.i. L^−1^). The herbicide was applied using a precision bench sprayer delivering 300 L ha^−1^ at a pressure of 215 kPa and speed of about 0.75 m s^−1^, with a boom equipped with three flat-fan (extended range) hydraulic nozzles (Teejet, 11002). Four weeks after herbicide application, the number of surviving plants (likely capable of producing seeds) and visual estimation of their biomass (VEB) were assessed and expressed as a percentage with respect to the untreated plants. The VEB scores, ranging from 100 (for plants not affected by the herbicide, equal to the untreated control) to 0 (when the plants were clearly dead), were given to each treated tray. Mean values and standard errors of the replicates were calculated, and a *t*-test (α < 0.05) was performed to identify significant differences between the two experiments.

### 4.3. Amaranthus Species Identification

To determine the *Amaranthus* species involved, a simplified botanical key specifically set up to recognize the most common weedy amaranths was used [8]. After the ALS treatment (Section 4.2), plants were grown until maturity and the inflorescences of about 10 plants per accession were collected and air-dried. For each accession, 15 pictures were taken: a full picture of the whole inflorescences and (with a digital microscope) 4 and 10 detailed pictures of single inflorescences and flowers plus seeds, respectively.

### 4.4. Resistance Mechanism

#### 4.4.1. DNA Extraction from Single Seeds

A protocol based on Tris-KCl-EDTA (TKE) [Tris–HCl 100 mM, pH 9.5, KCl 1 M, EDTA 10 mM] buffer, originally developed for plant tissue, had been adapted [33] to extract DNA from single DNA *Amaranthus* seeds. The protocol avoids the use of liquid nitrogen and toxic chemicals, plus it is carried out at room temperature and with no incubation time. Single seeds were ground with two bearing beads (4 mm diameter) in 2 mL vials by using Tissuelyser II (Qiagen, Hilden, Germany). Six hundred μL of extraction buffer were added after grinding, then beads were removed. Samples were then centrifuged for 5 min at 10,000 rpm to precipitate the debris. The aqueous phase of all samples was transferred into new vials and added with an equal volume of cold isopropanol. Vials were centrifuged at 10,000 rpm for 5 min and then supernatant was gently poured off to avoid losing the very scarce pellet at the bottom. Two hundred µL of 70% ethanol were used to wash pellets. Pellets were then allowed to air dry until complete ethanol evaporation and then re-suspended in 30 μL of water. DNA quality was determined with Nanodrop 2000c (Thermo Fisher Scientific, Waltham, MA, USA).

#### 4.4.2. ALS Partial Amplification and Sequencing

The genomic DNA was extracted from seeds of six accessions (namely, 20, 34, 42, 45, 49 and 56) approximately spanning the sampling area (three seeds each) and the *ALS* gene was partially amplified to look for point mutations at position 574. Primers AMA-2F (TCCCGGTTAAAATCATGCTC) and AMA-2R (CTTCTTCCATCACCCTCTGT) were used to amplify a region 337 bp following a previously described procedure [13] adapting volumes to seed-extracted gDNA. PCR was performed using GoTaq^®^ G2 Hot Start Polymerase (Promega, Madison, WI, USA) in a 50 µL mixture including 10 µL of 5X Green GoTaq Flexi Buffer, dNTPs mix (0.2 mM each), MgCl_2_ (1.5 mM), forward and reverse primers (0.2 µM), 0.25 µL GoTaq DNA Polymerase (5 U/µL) and 3 uL of seed-extracted gDNA (approximately 30 ng). Amplification conditions: 2 min at 95 °C; 30 cycles of 30 s at 95 °C, 30 s at 58 °C, 30 s at 72 °C; 5 min at 72 °C. Amplicons were purified using NucleoSpin^®^ Gel and PCR Clean-up kit (Macherey-Nagel GmbH & Co., Duren, Germany) following manufacturer’s instructions and Sanger sequenced (BMR Genomics, Padua, Italy).

#### 4.4.3. Allele-Specific PCR Assay (PASA) Development and Application

An allele-specific PCR assay (PASA) utilizes primers designed to selectively bind to only one allele [34] and is a widely adopted resistance detection technique [35]. Primer specificity is due to the presence of a nucleotide exactly matching one of the alleles but not the other. An allele-specific PCR assay was developed by using seed-extracted gDNA to rapidly detect the presence of the *ALS* mutant alleles leucine and methionine at position 574 among the accessions with survival rate to thifensulfuron-methyl higher than 25% (Section 4.2). Allele-specific primers (Table 1) were designed to recognize the wild type codon tryptophan and the two resistance-endowing allelic variants. To test the PASA specificity, three accessions described in the previous step (see Section 4.4.2) were used as reference samples positive for leucine, tryptophan or methionine, respectively; four samples each were amplified using all primer combinations, and then amplicons were visualized on 1% agarose gel. PASA were performed at similar conditions to that described for *ALS* amplification, with slight modifications: final volume 15 μL, DNA 1 μL (approximately 10 ng), annealing temperature 60 °C, elongation time 50 s. Once PASA specificity was verified, the presence of the two resistance-endowing alleles leucine and methionine was investigated across the selected populations (51). Genomic DNA was extracted from four seeds per accession as described in Section 4.4.1. All samples were amplified with two allele-specific PCR: one for the leucine allele and one for the methionine (204 PCR each, 408 in total). The PCR amplifications were run on 1% agarose gel: samples giving amplification with both primers were determined as positive for both leucine and methionine, while those giving no amplification were assumed to have no mutant alleles.

### 4.5. Susceptibility to Pre-Emergence Herbicides

To test if accessions were susceptible to non-ALS herbicides, the same six accessions described in Section 4.4.2 (namely, 20, 34, 42, 45, 49 and 56), plus the susceptible check 17–53, were treated with three non-ALS herbicides applied in pre-emergence. Herbicides were applied following a previously fine-tuned protocol [12], with the same sprayer used for ALS herbicide. Briefly, the pots were filled with a potting mix with no peat and covered with a layer of sterilized clay loam. Seeds were firstly vernalized in soaked paper and then germination was induced by exposing seeds to a single cycle of 28 °C for 12 h (day) and 18 °C for 12 h (night) in a germination cabinet. Twenty-four hours after sowing, pre-emergence herbicides were applied at the field rates: metribuzin (HRAC group 5/C1) 245 g a.i. ha^−1^ (Feinzin^®^ 70 DF, Adama, 70 g a.i. 100 g^−1^), clomazone (HRAC group 13/F4) 144 g a.i. ha^−1^ (Sirtaki^®^, Sipcam, 360 g a.i. 1000 mL^−1^) and metobromuron (HRAC group 5/C2) 1500 g a.i. ha^−1^ (Proman Flow^®^, Belchim Crop Protection, 500 g a.i. 1000 g^−1^). Two weeks after herbicides application, the number of seedlings and shoot fresh weight were recorded. The herbicide efficacy (E) was calculated as the percentage of 1- [survived seedlings in treated pots/seedlings in non-treated pots], while the relative fresh weight (FW) was calculated as the percentage of [the fresh weight of survived seedlings in treated pots/seedlings in non-treated pots]. The experimental layout was a complete randomized design with 3 replicates (pots) per treatment and 4 replicates per non-treated pots, and 50 seeds per pot. The experiment was conducted twice. Standard error (SE) was calculated for each data mean and data were analyzed with R [37] in RStudio [38]. A t-test (α < 0.05) was performed to identify significant differences between the two experiments.

## Figures and Tables

**Figure 1 plants-12-00332-f001:**
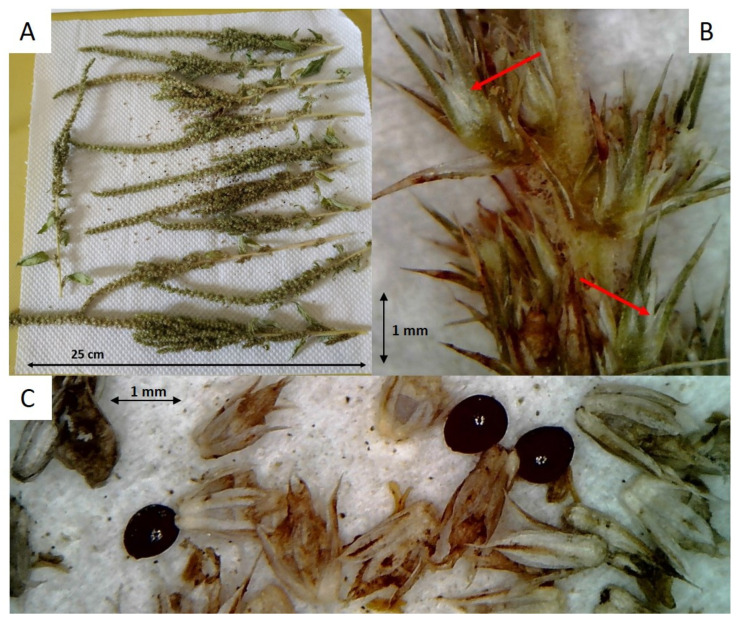
Example of the pictures taken for each of the 51 selected accessions: (**A**) whole inflorescences, (**B**) single inflorescences and (**C**) flowers and seeds. Red arrows indicate a membranous border that, in *A. hybridus*, stops approximately at half of the bract [8,15].

**Figure 2 plants-12-00332-f002:**
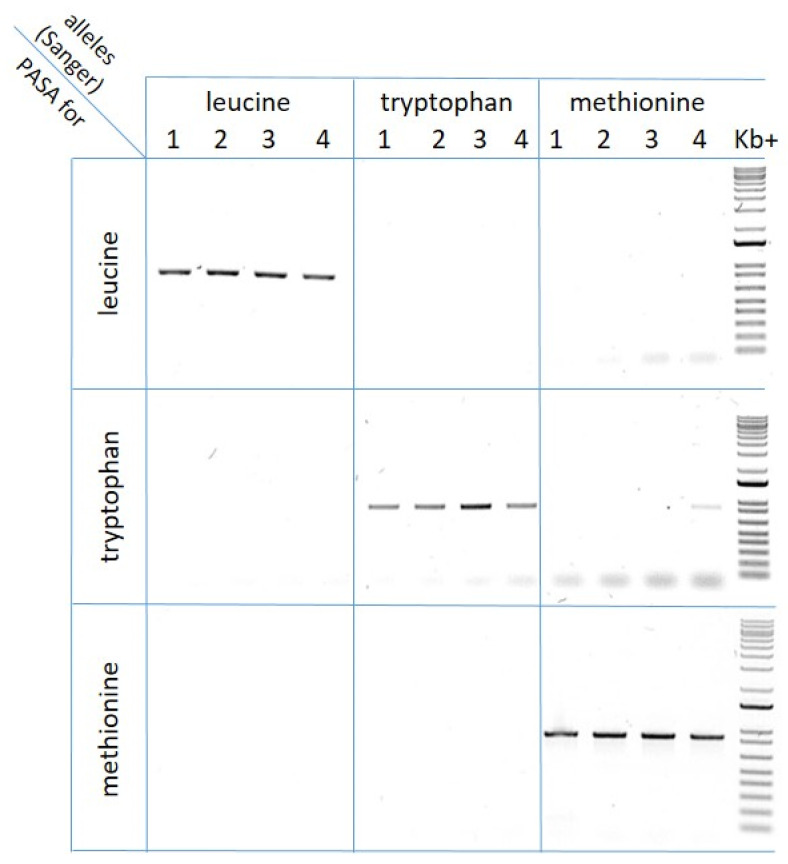
Setup of allele-specific PCR assay (PASA) using three PCR mixes specific to leucine (upper), tryptophan (middle) and methionine (bottom) on three *A. hybridus* reference seed samples: in these accessions, only the leucine, tryptophan or methionine was detected with Sanger sequencing (populations 42, 17–53 and 34, respectively), as described in Section 2.3.2.

**Figure 3 plants-12-00332-f003:**
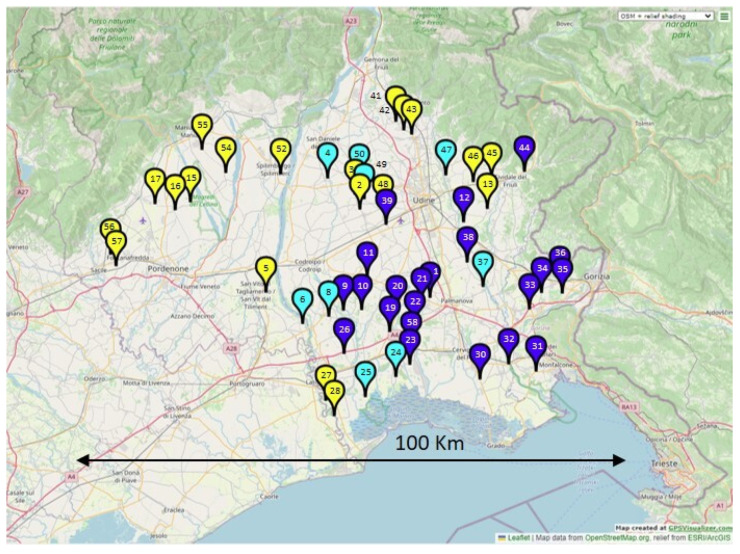
Geographical distribution of the accessions having only the leucine codon (yellow marker), those having only methionine (blue marker) and those having both alleles (cyan). The map was designed with the online tool GPS Visualizer [16].

**Figure 4 plants-12-00332-f004:**
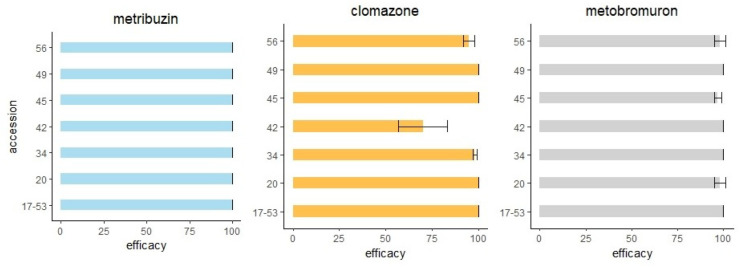
Mean values and standard errors are reported of metribuzin, clomazone and metobromuron efficacy, applied in pre-emergence. Accession 17–53 is the susceptible check.

**Figure 5 plants-12-00332-f005:**
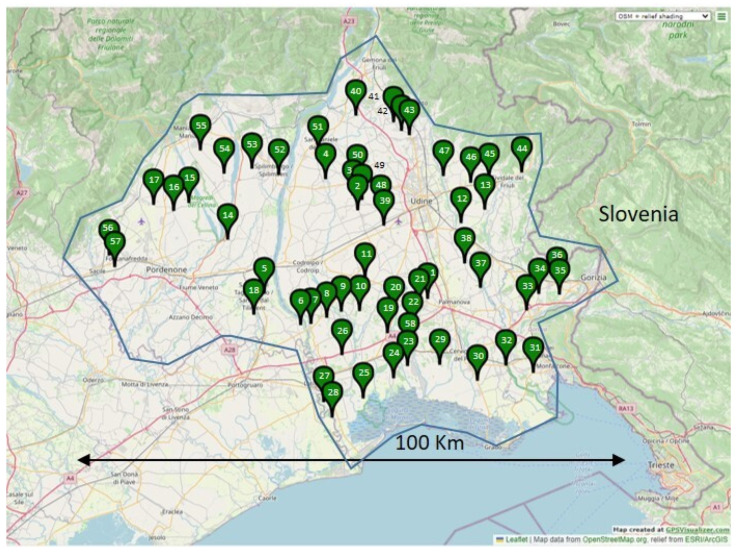
Sampling sites of the 58 *Amaranthus* spp. accessions. The blue line approximately represents the 3000 km^2^ area coved by the monitoring. Accession 58 was collected in 2012, while all the others in 2018. The susceptible check (17–53) is not visible as it was collected outside this area. The map was designed with the online tool GPS Visualizer [16].

**Table 1 plants-12-00332-t001:** Sequence of primers used for the allele-specific assay (PASA). An additional mismatch (A → C, underlined) was introduced to increase primer specificity, but at the third-to-last base of the primers instead of the penultimate base as elsewhere suggested [36].

for/rev	Primer Name	5′-3′ Sequence
Reverse	UTR3	TGGCTGATGAAAGGCAACAC
Forward	AS-Trp	ACATTTAGGTATGGTTGTTCAC**TG**
	AS-Leu	ACATTTAGGTATGGTTGTTCAC**TT**
	AS-Met	ACATTTAGGTATGGTTGTTCAC**AT**

## Data Availability

Main data are contained within the article or Appendix A. Biological material and GPS positions of accessions are available on request from the corresponding author due to privacy restrictions.

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
