# Peer review of "Diversity and Spread of Acetolactate Synthase Allelic Variants at Position 574 Endowing Resistance in Amaranthus hybridus in Italy"

_plants, 2023, doi:10.3390/plants12020332_

Round 1

Author Response

Thanks for the positive feedback. The point raised about the ploidy and breeding system (selfing rate) of Amaranthus hybridus has been added to the text, specifically all lines 322-324. As highlighted by the other reviewer, the results of Sanger and PASA obtained by seed-extracted DNA cannot be used to precisely determine the genotype, since both the embryo and the maternal tissues are extracted at the same time and so amplified. Therefore, the concept of homozygous and heterozygous samples, being imprecise, has been avoided across the whole text.

Reviewer 2 Report

This manuscript studies the diversity and spread of mutant ALS alleles endowing cross-resistance in Amaranthus hybridus populations of north-eastern Italy. The authors describe new protocols for seed DNA extraction and allele-specific PCR assay. I consider the manuscript is relevant and the writing is adequate.

I have some minor comments:

-Mutation at Trp574 was the only resistance mechanism evaluated. Non-target site resistance or other target-site mechanisms were not considered in this study. For this reason, specific objective c and some subtitles should be more precise (please see the attached document).

-There is a non-specific band in Fig. 3.

-Seed DNA extraction is useful for detecting resistant alleles in the populations but not for precise genotyping of individuals. Both embryo and maternal DNA samples are extracted in this protocol. I suggest changing sample and genotype designations (please see specific comments in the attached document).

-Other minor comments and suggestions are also included in the attached document.

Author Response

Thanks for the positive feedback. We agreed with most of the points raised by the reviewer. Here is a point by point answer to each of them.

L43-44, addressed, the newest herbicide classification released by HRAC has been added, now L43-44

L77, addressed, the aim has been changed  “c) to determine the main resistance mechanism”. An explanation on why no other resistance mechanism was investigated is now placed in the discussion (L326-331)

L82, addressed, “north-eastern Italy” has been added, now L83

L110, addressed, now L112

L115, we used Microsoft excel, but given the simplicity of the test and its routinary use, we think that this information is not relevant

L124, we think that our approach to determine the resistance mechanism had been “agnostic”, but data clearly indicated that no other resistance mechanism should be involved

L145, addressed, the PCR conditions were added, now L146-151

L200, addressed, R has been cited together with RStudio, now L204

L226, Figure 2, addressed, the meaning of the red arrows has been specified, L232-233

L231, same answer of L124

L240-241, addressed, the chromatogram picture depicting the three allelic variants has been added to supplementary material and referenced within the main body, now L247-248

L243, addressed, “seed sample” was used in place of “plants” when pertinent, e.g. L267-268

L250, Figure 3, addressed. We changed the designation as suggested

L250, Figure 3. Addressed. We added a phrase to explain that weak bands should be ignored because aspecific amplifications, now L254-256

L255, figure 3 caption. Addressed, the correct paragraph is 3.3.2 , L261

L275, Yes, results of the two repetitions were pooled. Addressed, a phrase was added, now L280-281

L290, addressed, we missed a reference, now L296-297

L293, addressed. The phrase has been shifted to the subsequent paragraph, now L299-301

L306-310, addressed. We do agree with the reviewer. The method is suitable to detect the presence/absence of specific alleles, but the genotype cannot be determined. We modified the main text accordingly to avoid the genotype concept, e.g. L252-259 or L262-267 and we better explained the limits and strength of the method, L310-324

L357, addressed, now L376

L444, addressed, now L482